# A Salt-Resistant Sodium Carboxymethyl Cellulose Modified by the Heterogeneous Process of Oleate Amide Quaternary Ammonium Salt

**DOI:** 10.3390/polym14225012

**Published:** 2022-11-18

**Authors:** Zhenfu Jia, Chengwei Zuo, Huishan Cai, Xiaojiang Li, Xiaodong Su, Jierui Yin, Wenlong Zhang

**Affiliations:** Department of Chemical Engineering, Chongqing University of Science and Technology, Chongqing 401331, China

**Keywords:** sodium carboxymethylcellulose, hydrophobic quaternary ammonium, salt resistance, polymer modification

## Abstract

In this study, hydrophobic quaternary ammonium intermediate was synthesized by epichlorohydrin (ECH) and oleamide propyl dimethyl tertiary amine (PKO). Sodium carboxymethylcellulose (CMC) was chemically modified by introducing a large number of hydrophobic quaternary ammonium branched chains to improve CMC’s salt resistance, thickening ability, and solubility. The quaternary ammonium salt structure can partially offset the compression double-layer effect of linear polymers in a low-price salt ion solution, which makes CMC more stretchable and helps it obtain a higher viscosity and greater drag-reduction performance. The experiment was mainly divided into three parts: Firstly, we performed an epichlorohydrin and oleic acid PKO reaction, generating an oleic acid chain quaternary ammonium chlorine atom intermediate. Secondly, the etherification reaction between intermediate –Cl and –OH groups of CMC was completed. Finally, the modified CMC was characterized by IR, SEM, and XPS, and the viscosity and the drag-reduction rate were evaluated. After CMC and the intermediate were reacted at a mass ratio of 9:1.8 at 80 °C for 5 h, the new CMC with enhanced thickening ability, salt resistance, and drag-reduction performance was obtained. We found that the apparent viscosity increased by 11%, the drag reduction rate increased by 3% on average, and the dissolution rate was also significantly accelerated, which was ascribed to the introduction of quaternary ammonium cation. Moreover, the oleic acid amide chain increased the repulsive force of the CMC chain to low-priced metal cations in solution and intermolecular repulsive force, which is beneficial to increase the viscosity, salt resistance, and drag-reduction performance.

## 1. Introduction

Due to fewer and fewer high-permeability conventional oil and gas fields, secondary or tertiary oil recovery needs to be completed in many oil fields [1]. Researchers pay more attention to the development of low-permeability and tight unconventional oil and gas fields. Hydraulic fracturing is significant in low-permeability reservoirs, where natural hydraulic fractures are minor and require a large amount of fluid with a lower density than the parent rock to expand the fractures. Thus, the proppant is introduced to broaden the fractures and facilitate the displacement of oil and gas from the rock [2]. However, water-based fracturing fluids are the most important part of hydraulic fracturing technology [3]. There are three main types of water-based fracturing fluids: synthetic polymers, natural plant gums, and Cellulose and derivatives. Water-skiing fracturing fluid with polymer as the primary agent is the most commonly used fracturing fluid for unconventional oil and gas field exploitation [4]. However, its long linear chain structure is complicated to diffuse under the extrusion of electrostatic repulsion between metal ions, resulting in its poor performance in the salinity water environment. In addition, artificial polymers tend to decompose at high temperatures, causing severe damage to reservoirs [5], and are difficult to biodegrade after flowback. Therefore, in practice, other thickeners are often mixed to neutralize defects.

Cellulose and its derivative units are composed of D-glucopyranose [6,7], a natural macromolecule compound, which is widely used in bacterial cultivation [8], aerogel production [9], and medical nano sterilization cotton [10]. Cellulose has good water solubility, ionic compression resistance, double electric layer effect, and temperature resistance, and the rheological property of Cellulose is also better than artificial polymers for reducing filtration loss [11]. The Cellulose-derivative fracturing fluid can effectively reduce the occurrence of water-locking effect in the application for sandstone with capillary pore structure and difficult reflux, and it also exhibits excellent temperature resistance and anti-swelling performance without excessive anti-swelling additives [12]. Therefore, Cellulose-based fracturing fluid is expected to partially replace artificial polymer fracturing fluid and become the new star in the unconventional oil and gas fracturing field. However, Cellulose fracturing fluids have serious disadvantages, such as a lower molecular weight than synthetic polymers, more metamorphic decomposition of the polysaccharide skeleton, and greater friction. So far, some functions of Cellulose are modified by chemical modification to improve these disadvantages [13,14]. In McCormick’s work [15], epichlorohydrin (ECH) reacted with –OH of CMC under acidic and heated conditions, and then Cellulose was easily modified by 1-chloro-2-hydroxypropane. Nishiyama et al. [16] found that hydrogen bonds between hydroxyl groups of Cellulose would change after alkaline swelling, reducing crystallinity or even recrystallizing, which significantly promoted modification such as an etherification reaction. Available Cellulose derivatives were produced based on this principle [17]. After modification of carboxymethyl hydroxyethyl Cellulose with a low degree of substitution by hexyl dimethylamine, the apparent viscosity increased by 54.6 mPas, and the thixotropy, viscoelasticity, and salt tolerance were all improved [18], which indicated that the properties of Cellulose derivatives could indeed be significantly improved by grafting long-chain hydrophobic groups. The insertion of quaternary ammonium salt functional groups into the structure of Cellulose can compensate for the negative charge density and increase the salt resistance of Cellulose [19]. Cellulose was dissolved in DMAC/LiCl solution. Through homogeneous reaction, P-Iodobenzoic acid Cellulose can be synthesized by esterification of Cellulose and Iodobenzoyl chloride, and the highest degree of substitution (DS) can reach 2.05 [20]. The samples with a DS around 1 have good viscoelasticity and salt tolerance and a good dissolution rate. Sodium carboxymethyl Cellulose will partially open the crystallization zone and expose a certain amount of active OH structure under alkaline swelling and high-temperature conditions, and heterogeneous dehydration reactions with amides can take place by using this mechanism [21].

In order to obtain Cellulose-fracturing fluid thickener with salt resistance and high drag reduction, some modified mechanisms and methods have been proposed so far. Hydrophobic quaternary ammonium chains were synthesized by ECH and PKO oleate under acidic conditions and modified by heterogeneous derivatizing with CMC. However, most current experiments have focused on the homogeneous modification of Cellulose derivatives. There are few reports on the heterogeneous modification of Cellulose derivatives with a high degree of substitution. In this work, the Cellulose derivatives with substitution degree were modified by the heterogeneous process of oleate amide quaternary ammonium salt, and the modified sodium carboxymethyl Cellulose exhibited better salt-resistance.

## 2. Experimental Materials and Methods

### 2.1. Material

CMC (industrial grade) with a degree of substitution of 0.83 was purchased from Chongqing Lihong Fine Chemical Co., LTD in Chongqing, China. Epichlorohydrin, distilled water, hydrochloric acid, DMF solution, NaOH solid, and ethyl acetate were purchased from Chengdu Colon Chemical Co., LTD. Oleate amide propyl dimethyl tertiary amine (Analytical purity) was purchased from Nantong Shajia Chemical Technology Co., LTD in Nantong, China.

### 2.2. Experimental Methods

#### 2.2.1. Synthesis of Hydrophobic Chain Intermediates of Quaternary Ammonium Oleate

Oleic acid hydrophobic quaternary ammonium chain intermediate was prepared by oleic acid amide propyl dimethylamine (PKO) and ECH reaction. The experimental scheme was as follows: PKO oleic acid and concentrated hydrochloric acid were added to the 250 mL flask, under stirring, for 10 min, at room temperature. After mixing evenly, we added the 10.17 g ECH slowly, and the reacted temperature was set at 60 °C, under stirring for 2 h. The reaction equation is shown in Figure 1.

The reaction using the epoxide group itself is livelier. It is easy in the proton acid environment for H^+^ protons to produce a proton ether ring with a positive charge on the oxygen atoms; weaken the C–O keys to carbon atoms with positive charge at the same time; increase the ability to combine with nucleophilic reagent; and, again, in the tertiary amine structure of nucleophilic reaction, ECH can thus be connected to the oleic acid PKO long-chain quaternary ammonium groups. The reaction equation is as Figure 2:

After the reaction was completed, ethanol and excess ECH were evaporated by rotary evaporators at 60 °C. The excess ethyl acetate was added to a flask, the flask mouth was closed, and the flask was recrystallized in a refrigerator at −18 °C for 8–12 h. After removal, the supernatant fluid was quickly decanted, ethyl acetate was added, and the above steps were repeated three times to obtain the long-chain hydrophobic quaternary ammonium salt intermediate with purity of more than 90%.

The ^1^HNMR characterization and analysis of hydrophobic quaternary ammonium chain intermediates are shown in Appendix A.

#### 2.2.2. Modification of the CMC

The 0.01 mol/L NaOH solution was prepared first. Then 50 mL DMF solvent was set in a 250 mL flask, and a 1 g CMC powder was added to the flask under magnetic stirring, obtaining a suspension system. Next, a total of 5 × 10^−4^ mol of NaOH was dropped into the flask in solution form and magnetically stirred for 10 min. Subsequently, some quaternary ammonium intermediate was added to the flask and heated at 80 °C, with magnetic stirring, for 5 h.

The reaction mechanism is that chlorine atoms in long-chain molecules of hydrophobic quaternary ammonium salt are activated under an alkaline environment, resulting in strong nucleophilicity. At the same time, CMC expands in alkaline environment and partially opens the crystallization zone, so that NaOH attacks –OH in the crystallization zone and generates CMC–ONa. Subsequently, the sodium alcohol group of CMC and the Cl atom in the PKO–ECH intermediate [22] undergo Williams reaction to generate CMC–O–PKO–ECH. The reaction equation is as Figure 3:

Due to the heterogeneous reaction, the solid–liquid separation state was observed after the completion of the reaction. The flask was removed from the oil bath pot, the excess DMF solvent was poured out, excessive ethyl acetate was added, and the supernatant was then poured out after the layering; the steps were repeated three times, and then the new CMC powder with high purity was obtained by rotary evaporation at 60 °C.

### 2.3. Characterization Analysis

#### 2.3.1. Characterization of Rheology

Anton Paar rheometer model MCR102 was provided by Anton Paar GmbH (Graz, Austria) and was used to test the storage and loss modulus of the CMC crosslinking liquid.

#### 2.3.2. Characterization of IR

Fourier infrared spectrometer model tension-27 was provided by Germany Brock Co. LTD in Bochum, Germany. It was used to detect the changes of the functional groups of CMC before and after modification and to characterize the functional groups of intermediates.

#### 2.3.3. SEM Scanning

SEM electron microscope model FEI was provided by Inspect F50, from the US. The pretreatment method of this scanning electron microscope is as follows: CMC powder is dissolved in distilled water to make a solution, lyophilized for a certain time, and then sublimated into solid at low temperature for scanning. The advantages of this method are that the fluid morphology of CMC in aqueous solution can be obtained, which is of greater significance for the analysis of CMC in practical use, and it is more convenient to observe the comparison of structure and solubility before and after modification. Before and after modification, it is more convenient to observe the comparison of micro morphology.

#### 2.3.4. Drag Reduction Rate Characterization

Fracturing fluid pipeline friction meter HMAZ-IV was provided Jiangsu Hua ’an Scientific Research Instrument Co., LTD in Jiangsu, China. The measurement method is to slowly add 3 g or 5 g CMC powder into a liquid storage tank containing 10 L of pure water, under a state of agitation. Then calculate the velocity of fluid with the following Formula (1):(1)$$f\left(\%\right)=\frac{Pa1-Pa2}{Pa1}$$
where *f (%)* is the drag reduction rate, *Pa*1 is pure water pressure difference, and *Pa*2 is powder base fluid pressure difference.

#### 2.3.5. XPS Characterization

The XPS model ESCALAB Xi^+^ was provided by Thermo Scientific (Waltham, MA, USA). It was used to measure the change of N element content and C energy level of CMC before and after modification.

#### 2.3.6. HNMR

Nuclear magnetic resonance (NMR) model Brooker 400 M was provided by Germany Brock Co. LTD in Bochum, Germany. It was used to measure the H types of hydrophobic quaternary ammonium intermediates to determine the successful synthesis of intermediates.

## 3. Results and Characterization

### 3.1. Preparation of Modified CMC

In order to optimize the reaction conditions for modifying CMC, a set of experimental schemes was designed, as shown in Table 1.

(1)Influence of mass ratio of intermediate to CMC on product viscosity

In order to analyze the influence of the mass ratio of CMC and intermediate for the effect of the modification product, the mass ratio of CMC and hydrophobic quaternary ammonium intermediate was controlled between 1.5:9 and 1.9:9. Moreover, we fixed the reaction temperature, reaction time, and the amount of NaOH solution. The resulting apparent viscosity changes are shown in Figure 1.

The modified CMC exhibited the better apparent viscosity in NaCl, indicating the better salt resistance in NaCl compared to CaCl_2_. When the mass ratio is 1.8:9, the apparent viscosity changes dramatically; the viscosifying ability effect is greatly improved, and the viscosity is higher than that of the other three proportions. The reason may be that, when the mass ratio between the intermediate and CMC is very small, the reaction is not sufficient and the number of groups on the modification is small. However, when the mass ratio is high, a large number of intermediates produce a steric effect, which hinders the etherification reaction with CMC, resulting in poor results.

(2)The influence of reaction time

By fixing the mass ratio, reaction temperature, and the amount of NaOH solution, the apparent viscosity was obtained by controlling different reaction times to determine the optimal reaction time, as shown in Figure 2. With an increasing reaction time, the apparent viscosity of the generated CMC powder increases first and then decreases, which is the lower modification efficiency and the smaller contact site probability caused by insufficient reaction. If the reaction time is too long, chain-breaking hydrolysis will occur at a high temperature, reducing the molecular weight of CMC.

(3)The influence of reaction temperature

As shown in Figure 3, the apparent viscosity of the generated CMC increased rapidly between 75 °C and 80 °C, where the reaction temperature is low, the modification efficiency is low, and the reaction is difficult to occur. Then the apparent viscosity of the generated CMC was decreased at 85 °C, whereas the hydrolysis speed of CMC was accelerated at a high temperature, causing the destruction of structure.

(4)The effect of the amount of NaOH solution

As shown in Figure 4, when the 5 × 10^−4^ mol NaOH was added, the apparent viscosity of the modified CMC reached the maximum value. The balance between swelling and hydrolysis was obtained only at a reasonable NaOH range, and part of the crystallization zone of Cellulose reacted with hydroxyl group into sodium alcohol, which promotes the occurrence of an etherification reaction. If the NaOH is not enough, it is not enough to open the crystallization zone for a reaction. If the alkalinity is too strong, the hydrolysis reaction of Cellulose is accelerated, resulting in the destruction of molecular structure and the decrease of apparent viscosity.

When CMC is connected to hydrophobic chains, the association effect between molecules and within molecules will occur after it is dissolved in water, and the macromolecular chains will aggregate to form supramolecular structure, which increases the volume of molecular hydrodynamics and the viscosity of solution. The diagram is shown in Figure 5.

### 3.2. Characterization Results of Modified CMC

#### 3.2.1. Infrared Spectrogram

We compared the infrared spectra of the CMC before and after modification, as shown in Figure 6, and found that the modified CMC increases an absorption peak at 1592 cm^−1^, which can be attributed to the bending vibration peak (in-plane) of ammonium NH. The absorption peak at 3432–3424 cm^−1^ is the contraction vibration peak of hydroxyl O–H. The absorption peaks at 2920 cm^−1^ and 2850 cm^−1^ are vibration absorption peaks of methyl C–H. The absorption peaks at 1350 cm^−1^, 1386 cm^−1^, and 1339 cm^−1^ are the contraction vibration peaks of C–O, and 1459 cm^−1^ is a new peak after modification [23], confirming that it is a C–N characteristic peak. Therefore, the hydrophobic chain of quaternary oleamide ammonium was successfully grafted onto CMC.

#### 3.2.2. SEM

The unmodified and modified electron microscopy images are presented in Figure 7. Figure 7a shows the 200 μm electron microscopy of the unmodified CMC, and Figure 7b shows the 200 μm electron microscopy of the modified CMC. As shown in the Figure 7, the CMC before modification presented a dense network structure, while the CMC after modification presented a filamentous sheet structure. This is because the modified CMC opens the crystallization zone to a certain extent, weakens the bonding effect of hydrogen bonds on CMC, and makes the molecular structure stretch, which is more conducive to the dissolution of CMC in water. Figure 7c,d show the images of CMC under 50 μm electron microscope before and after modification, respectively. It can be clearly seen that, before modification, the meridians have a simple structure, clear skeleton veins, and small cracks, but after modification, they become filaments and contain dense long chains, and the main trunk has no cracks, thus proving that, after modification, the solubility is increased, and the long chain of hydrophobic quaternary ammonium salt was connected to this surface. Figure 7e,f show the images under a 20 μm electron microscope. It is more obvious that there are many cracks in the trunk before modification, and these cracks result from the poor solubility of CMC in water due to hydrogen bonding; the solution is also brittle. There are no cracks in the trunk of CMC after modification, and the branch chain structure obviously attaches to the trunk after modification and presents a filamentous suspension. Figure 7g,h are the images under the electron microscope of 10 μm, which is not different from the detail of 20 μm but offers more clarity. The eight electron microscopy images a–h show that the modified CMC, indeed, has better ductility and solubility, and a large number of hydrophobic quaternary ammonium salt chains were grafted on the surface.

#### 3.2.3. XPS

According to the CMC–O–PKO–ECH all-element scanning results in Figure 8, the content of C accounted for 77.2%, the content of O accounted for 19.31%, and the content of N accounted for 3.49%. Figure 9 shows CMC’s before-modification all-element scanning results’ the content of C accounted for 55.68%, the content of O accounted for 43.36%, and the content of N accounted for 0.97%.

In addition, observing the C1s energy spectrum of Figure 10 shows that there are four energy spectrum peaks, A, B, C and D, among which the low binding energy between the C–H and C–C bonds is about 285–284.5 eV, the energy spectrum peak between C and non-carbonyl O is about 286.5 eV, and the B peak is about 286.5 eV. When C is connected to carbonyl O, the oxidation state is higher, and the electron binding energy is larger. The energy spectrum peak is about 288 eV, namely the C peak [24,25]. Finally, when C is connected to N, the binding energy reaches the maximum, generally up to about 289 eV, namely the D peak.

According to Figure 11, there are only three C1s energy peaks for unmodified CMC, and they coincide with peaks A, B, and C in Figure 10. However, when comparing Figure 10 and Figure 11, we can clearly see the difference of C energy peak in 289 eV, and the contents of C, O, and N of the CMC before and after modification were changed; as can be seen, the hydrophobic quaternary ammonium branched chain was successfully attached with a high grafting rate and attached to the surface of CMC crystal.

### 3.3. Performance Evaluation of Modified and Unmodified CMC

#### 3.3.1. Apparent Viscosity

Figure 12 shows a comparison of the apparent viscosity of CMC dissolved in pure water before and after modification. It was found that the viscosity of CMC increased by 11% after modification. the introduction of hydrophobic molecular chains increased the molecular weight of CMC and the association ability between CMC and water, thus intensifying the hydromechanical volume and allowing us to obtain excellent rheological properties.

Figure 13 shows the apparent viscosity comparison of CMC dissolved in 10,000 ppm NaCl solution before and after modification. With the increase of CMC concentration, the apparent viscosity can be stably maintained more than 10% higher than that before modification, indicating that the modified CMC has better rheological properties in NaCl aqueous solution. We proved that the modified quaternary ammonium salt has a better effect in regard to resisting the cationic compressed electric layer.

Figure 14 shows the apparent viscosity comparison of CMC dissolved in 30,000 ppm NaCl solution before and after modification. Compared to the CMC dissolved in pure water (Figure 12) and in 10,000 ppm (Figure 13), the viscosity of the unmodified and modified CMC was decreased with an increasing Na^+^ salt concentration. With the increase of Na^+^ content, the compression effect on CMC molecular chain also increases, which greatly reduces the hydromechanics volume of CMC. However, it can be seen from the apparent viscosity that the modified CMC has better salt tolerance. When the concentration of CMC increases from 0.1% to 0.5%, the apparent viscosity of the modified CMC increases by more than 10% compared to before, thus proving that the modified CMC still has good rheology in the high-Na salt environment and can achieve the purpose of using the high salinity formation flowback liquid confect solution.

Figure 15 shows the apparent viscosity comparison of CMC dissolved in 10,000 PPM CaCl_2_ solution before and after modification. We can see that both before the modification and after modification of the CMC, the viscosity has a great drop; this will be reflected in any other polymeric thickener because the Ca^2+^ in the water volume of the polymer fluid mechanics has a great compression effect, making it so that the long-chain molecules cannot stretch and then lose most of the viscosity. However, after quaternization, CMC still has a good resistance to Ca^2+^, and its apparent viscosity is nearly 20% higher than that of unmodified CMC, which is also of great significance in practice.

#### 3.3.2. Drag Reduction Rates

Figure 16 shows the drag-reduction-ratio comparison between modified CMC and unmodified CMC in the pipeline friction meter. As can be seen from the bar chart, the drag-reduction rate changes by about 2–3% before and after modification, and the improvement is not obvious. Due to the low molecular weight of the CMC itself, the overall molecular weight of the unit cannot be changed, and only modified in a small amount of hydrophobic chain quaternary ammonium salt. however, with the increasing number of branched chain, the rheology can be enhanced as the flow speed and drag reduction rate increase gradually to increase. Thus, the modified long-chain molecules can effectively create kinetic energy and elastic potential energy low-loss convert each other, inhibiting the development of turbulence.

Similarly, Figure 17 shows the drag-reduction data of CMC before and after modification at 0.05% concentration. With the increase of concentration, the drag-reduction effect increases correspondingly, but the difference with 0.03% concentration does not increase significantly, indicating that the drag-reduction effect stably increases by 2–3%.

#### 3.3.3. Comparison of Rheological Properties before and after Modification

It can be seen from Figure 18, which shows the storage and loss modulus of the Anton Paar rheometer, that the elastic modulus of unmodified CMC decreases with the increase of angular velocity shear rate after the crosslinking of organozirconium crosslinker in aqueous solution. In contrast, the loss modulus begins to rise with the growth of angular velocity, and the loss modulus and storage modulus tend to cross. This is because unmodified CMC only has carboxyl group crosslinking sites with better crosslinking effects, while unmodified CMC has fewer hydroxyl groups and poor crosslinking effects. With the increase of the shear rate, the viscoelastic energy gradually weakened, the elastic gel began to thin, and the sand-carrying capacity weakened. After the modified CMC was dissolved in water and became an elastic gel, the elastic modulus remained stable with the increase of the shear rate, while the viscous modulus grew slowly and the cross trend was not obvious, indicating that the viscoelasticity was greatly improved. This is because the CMC of the intramolecular force is abated by modification, making molecules more stretch and exposing more of the hydroxyl, and the hydrophobic chain quaternary ammonium is adhered to the hydroxyl, to some extent, increasing the repulsion between the molecular weight and molecular, thus forming a more stable intermolecular crosslink network. Thus, when the shear is not easy to deform, the viscoelasticity is significantly increased, the carrier better, and there is a more stable performance [26].

## 4. Conclusions

Here, the PKO–ECH hydrophobic quaternary ammonium oleate intermediate was connected to CMC by etherification reaction under alkaline conditions, and its properties in regard to its pure water viscosity, salt resistance, drag reduction, storage modulus, and loss modulus were improved. This method is suitable for the secondary modification of Cellulose derivatives with high substitution degree and high performance, and the performance can be greatly improved while the framework structure is basically maintained. The surface of the crystallization zone attached to the modified hydrophobic group structure can significantly improve the derivatives’ rheological properties and salt tolerance. In the future, related articles will be published about the graft copolymerization of CMC with acrylamide and small molecular functional monomers by homogeneous derivation method to prepare high-molecular biological thickening-agent polymers with a stronger resistance reduction rate and greater salt tolerance.

## Data Availability

Not applicable.

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
