# Peer review of "A Salt-Resistant Sodium Carboxymethyl Cellulose Modified by the Heterogeneous Process of Oleate Amide Quaternary Ammonium Salt"

_polymers, 2022, doi:10.3390/polym14225012_

Round 1

Reviewer 1 Report

The manuscript entitled ‘A salt-resistant sodium carboxymethyl cellulose modified by the heterogeneous process of oleate amide-quaternary ammonium salt’ contains the synthesis, and the rheological studies of modifies CMC with a  hydrophobic quaternary ammonium oleate intermediate.

I think it is desirable to reconsider it after addressing the following minor and major issues.

1)      Change the captions of the chemical reactions to Figure of Scheme.

2)      The English need improvement. i.e ‘Reaction steps: Prepare 0.01mol/L NaOH solution in advance, then add 50mlDMF solution into a 250ml flask, drop 5mL of the prepared NaOH solution, add the magnet rotor, put it into the oil bath magnetic stirring pot, adjust the speed to 100rpm, stir and add CMC. After swelling at room temperature for 5min, the intermediate was added and the reaction was increased to 80℃ for 5h’ this is not a scientific writing.

3)      The quality of all figures must be improved.

4)      There are a lot of typos throughout the text. i.e line 161 the ?

5)      The presented ratios are between CMC and what?

6)      In figure 2, the reaction time is measured in sec, min, hours?

7)      In figure 3, the same what is the unit of temperature?

8)      Please change the amount of NaOH to moles in figure 4 instead of ml.

9)      Please add a table with the ratios/temperatures/NaOH etc

10)   The synthesized intermediates as well as the modified CMC must be characterized with NMR to verify the successful synthesis.

11)   Please consider modifying the reference section according to the journal’s format.

12)   The manuscript is confusing. The English need a lot of improvement. Please work on that.

Author Response

Dear reviewing professors:

We greatly appreciate for your hard work on this manuscript titled A salt-resistant sodium carboxymethyl cellulose modified by the heterogeneous process of oleate amide-quaternary ammonium salt (Manuscript ID: polymers-2016028). We have considered the reviewers’ comments and realized that some significant and necessary revision should be made to clarify our points. Parts of abstract, introduction, experimental, results and discussion are all rechecked to improve readability of manuscript and to make it easier for scientists to find what they are interested.

The following are the point-by-point responses to the concerns from reviewers. Thank all the reviewers for their good suggestions and affirmation of some interesting points in this manuscript, which encouraged us a lot.

Thanks again for your patience and kindness.

With Best Regards!

Yours Sincerely,

Zhenfu Jia

Reviewer(s)' Comments to Author:

Reviewer: 1

Comments to the Author

The manuscript entitled ‘A salt-resistant sodium carboxymethyl cellulose modified by the heterogeneous process of oleate amide-quaternary ammonium salt’ contains the synthesis, and the rheological studies of modifies CMC with a hydrophobic quaternary ammonium oleate intermediate.

Comment 1. Change the captions of the chemical reactions to Figure of Scheme.

Answer: We are appreciated your comments on this work. After realizing this loophole, we actively made corrections and changed the three reaction equations into figure of scheme.

Comment 2.  The English need improvement. i.e ‘Reaction steps: Prepare 0.01mol/L NaOH solution in advance, then add 50mlDMF solution into a 250ml flask, drop 5mL of the prepared NaOH solution, add the magnet rotor, put it into the oil bath magnetic stirring pot, adjust the speed to 100rpm, stir and add CMC. After swelling at room temperature for 5min, the intermediate was added and the reaction was increased to 80℃ for 5h’ this is not a scientific writing.

Answer: Thank you again for this crucial advice, which has greatly improved my writing. After I saw your opinion, I immediately made positive rectification, and now it has been changed to: ‘The 0.01mol/L NaOH solution was prepared firstly. 50mL DMF solvent was set in a 250ml flask, and a 1g CMC powder was added into the flask under magnetic stirring, obtaining a suspension system. Then a total of 5x10-4mol of NaOH was dropped into the flask in solution form and magnetically stirred for 10 minutes. Subsequently, some quaternary ammonium intermediate was added into the flask, and heated at 80 ℃ with magnetic stirring for 5h’.

Comment 3, 6, 7 and 8. The quality of all figures must be improved. In figure 2, the reaction time is measured in sec, min, hours? In figure 3, the same what is the unit of temperature? Please change the amount of NaOH to moles in figure 4 instead of ml.

Answer: Thanks for your suggestion. I think ideas 3,6,7 and 8 are of the same kind, so I combine them together as a reply. First, I made all the images clear, bolded the numbers and titles, and optimized the fonts. Then, units (h) were added to Figure 2 and units (℃) to Figure 3, and the unit of NaOH was changed to (mol) in Figure 4.

Comment 4. There are a lot of typos throughout the text. i. e line 161 the?

Answer: I'm terribly sorry. The missing words here have been made up.

Comment 5. The presented ratios are between CMC and what? 

Answer: Is the ‘mass ratio between CMC and oleamide hydrophobic quaternary ammonium salt intermediate’ has been filled

Comment 6. Please add a table with the ratios/temperatures/NaOH etc

Answer: Thanks for your suggestion. This table has been drawn by me and put into ‘3.1Preparation of modified CMC’ on page 9 of the article.

Comment 7. The synthesized intermediates as well as the modified CMC must be characterized with NMR to verify the successful synthesis

Answer: I must apologize to you for this problem and explain it clearly. First of all, the NMR hydrogen spectrum of the intermediate has been given in Appendix 1, Fig. S1. However, due to the poor solubility of the deuterium reagent, the NMR hydrogen spectrum of CMC made by repeated mixing for more than 30 times is still not ideal, and the peak pattern is not clear, which cannot form an effective comparison relationship. However, we made a supplementary XPS characterization of unmodified CMC. The C-level peak and N element content of CMC before and after modification were compared, and the obvious difference can be seen. Together with the clear graph comparison of scanning electron microscopy, we believe that the modification can be proved successful

Comment 8. Please consider modifying the reference section according to the journal’s format.

Answer: The format of the references section has been changed to the required format for ‘DMPI’.

Comment 9. The manuscript is confusing. The English need a lot of improvement. Please work on that.

Answer: Thank you again for your suggestion, which is of great help to my promotion and progress. We attach great importance to it and have made positive modifications. Many words, grammar and voice in the whole text have been corrected. Please review it again.

Thank you again.

Reviewer 2 Report

This manuscript described the modification of CMC using hydrophobic quaternary ammonium intermediate to improve the salt resistance and drag reduction performance of the finished grade CMC.

In my opinion, the salt resistance and drag reduction performance of modified CMC are not that effectively improved compared with unmodified CMC. At the same time, this work showed no significant advantage compared to to Ref.16 (Glasser W G, Atalla R H, Blackwell J, Cellulose, 2012) and Ref.17(Es-said A, El Moussaouiti M, Bchitou R J. Polym. Res., 2019.). Moreover, the XPS and IR statistics of unmodified CMC were lacked, so the comparison is poor, and the explanation of the corresponding experimental phenomenon is not clearly and definitely. All those issues need to be carefully clarified. After that, this paper still needs major revision before acceptance

Other minor issues:

1.     The outstanding advantage of this work compared with the previous work should be proposed.

2.     The data in the whole article are very scattered, so it is better to integrate the data with comparison relationship, so that readers can more intuitively see the difference in performance of unmodified CMC and modified CMC.

3.     The authors could add the following references which would again increase the interest to general functional cellulosic material readers: Journal of Bioresources and Bioproducts, 2021, 6(1): 26-32; ACS Applied Materials & Interfaces, 2021, 13, 7617-7624;  Journal of Bioresources and Bioproducts, 2021, 6(1): 75-81.

Author Response

Dear reviewing professor:

We greatly appreciate for your hard work on this manuscript titled A salt-resistant sodium carboxymethyl cellulose modified by the heterogeneous process of oleate amide-quaternary ammonium salt (Manuscript ID: polymers-2016028). We have considered the reviewers’ comments and realized that some significant and necessary revision should be made to clarify our points. Parts of abstract, introduction, experimental, results and discussion are all rechecked to improve readability of manuscript and to make it easier for scientists to find what they are interested.

The following are the point-by-point responses to the concerns from reviewers. Thank all the reviewers for their good suggestions and affirmation of some interesting points in this manuscript, which encouraged us a lot.

Thanks again for your patience and kindness.

With Best Regards!

Yours Sincerely,

Zhenfu Jia

Reviewer: 2

Comments to the Author

This manuscript described the modification of CMC using hydrophobic quaternary ammonium intermediate to improve the salt resistance and drag reduction performance of the finished grade CMC.

In my opinion, the salt resistance and drag reduction performance of modified CMC are not that effectively improved compared with unmodified CMC. At the same time, this work showed no significant advantage compared to to Ref.16 (Glasser W G, Atalla R H, Blackwell J, Cellulose, 2012) and Ref.17(Es-said A, El Moussaouiti M, Bchitou R J. Polym. Res., 2019.). Moreover, the XPS and IR statistics of unmodified CMC were lacked, so the comparison is poor, and the explanation of the corresponding experimental phenomenon is not clearly and definitely. All those issues need to be carefully clarified. After that, this paper still needs major revision before acceptance.

Comment 1. The outstanding advantage of this work compared with the previous work should be proposed.

Answer: Thanks for your suggestion. Indeed, the improvement of viscosity and drag reduction by our modification is not significant, which is not as obvious as that in some literatures. However, the modified material we selected is a mature high-performance CMC, which has been screened out through layers of layers and already has more salt resistance and drag reduction ability than ordinary CMC, which makes it more difficult to improve its performance. In addition, our reaction is heterogeneous derivative reaction. The advantages of this reaction are that the reaction process is simple, the processing process is simple, the product purity is high, the structure is not easy to destroy, but the corresponding defect is that the degree of substitution is low, cannot greatly change the property, suitable for the secondary promotion of this kind of high-performance products.

Comment 2. The data in the whole article are very scattered, so it is better to integrate the data with comparison relationship, so that readers can more intuitively see the difference in performance of unmodified CMC and modified CMC.

Answer: Thank you for your comments. In my opinion, attention should be paid to this. For this reason, in 3.3'Performance evaluation of modified and unmodified CMC', we drew six comparison graphs of viscosity and drag reduction rate of CMC before and after modification. In addition, the loss modulus and storage modulus graphs of CMC before and after modification were combined for comparison in 3.3.3 section.

Comment 3. The authors could add the following references which would again increase the interest to general functional cellulosic material readers: Journal of Bioresources and Bioproducts, 2021, 6(1): 26-32; ACS Applied Materials & Interfaces, 2021, 13, 7617-7624; Journal of Bioresources and Bioproducts, 2021, 6(1): 75-81. 

Answer: Thank you again for your suggestion. We have accepted it and after careful reading, we have added the three articles you recommended to the introduction. The serial numbers are [8],[9] and [10] respectively. 

Comment 4. In your Comments to the Author, you mentioned that it is necessary to compare the XPS effect of CMC before and after modification and show the IR scanning results of hydrophobic quaternary ammonium intermediates

Answer: This suggestion is very important. We carefully made up the XPS of unmodified CMC and added the IR scanning results of monomer. We also made some modifications to the original text and added the comparison results into the manuscript.

Thank you again. 

Round 2

Reviewer 1 Report

The authors addressed the requested points. I am happy to recomment this manuscript to be published in Polymers in the present form.

Reviewer 2 Report

I think the revised version seems OK for me